# Look Ma, No Hands!
# Agent-Environment Factorization of Egocentric Videos

**Matthew Chang    Aditya Prakash    Saurabh Gupta**
University of Illinois, Urbana-Champaign
`{mc48, adityap9, saurabhg}@illinois.edu`

## Abstract

The analysis and use of egocentric videos for robotic tasks is made challenging by occlusion due to the hand and the visual mismatch between the human hand and a robot end-effector. In this sense, the human hand presents a nuisance. However, often hands also provide a valuable signal, *e.g.* the hand pose may suggest what kind of object is being held. In this work, we propose to extract a factored representation of the scene that separates the agent (human hand) and the environment. This alleviates both occlusion and mismatch while preserving the signal, thereby easing the design of models for downstream robotics tasks. At the heart of this factorization is our proposed Video Inpainting via Diffusion Model (VIDM) that leverages both a prior on real-world images (through a large-scale pre-trained diffusion model) and the appearance of the object in earlier frames of the video (through attention). Our experiments demonstrate the effectiveness of VIDM at improving inpainting quality on egocentric videos and the power of our factored representation for numerous tasks: object detection, 3D reconstruction of manipulated objects, and learning of reward functions, policies, and affordances from videos.

## 1   Introduction

Observations of humans interacting with their environments, as in egocentric video datasets [12, 20], hold the potential to scale up robotic policy learning. Such videos offer the possibility of learning affordances [2, 19], reward functions [1] and object trajectories [57]. However, a key bottleneck in these applications is the mismatch in the visual appearance of the robot and human hand, and the occlusion caused by the hand.

Human hands can often be a nuisance. They occlude objects of interaction and induce a domain gap between the data available for learning (egocentric videos) and the data seen by the robot at execution time. Consequently, past work has focused on removing hands from the scene by masking [19] or inpainting [1]. However, hands also provide a valuable signal for learning. The hand pose may reveal object affordances, and the approach of the hand toward objects can define dense reward functions for learning policies.

In this work, we propose the use of a *factored agent and environment representation*. The agent representation is obtained by segmenting out the hand, while the environment representation is obtained by inpainting the hand out of the image (Fig. 1). We argue that such a factored representation removes the nuisance, but at the same time preserves the signal for learning. Furthermore, the factorization allows independent manipulation of the representation as necessary. *E.g.*, the agent representation could be converted to a form that is agnostic to the embodiment for better transfer across agents. This enables applications in 2D/3D visual perception and robot learning (see Fig. 2).

---

Project website with code, video, and models: `https://matthewchang.github.io/vidm`.

37th Conference on Neural Information Processing Systems (NeurIPS 2023).

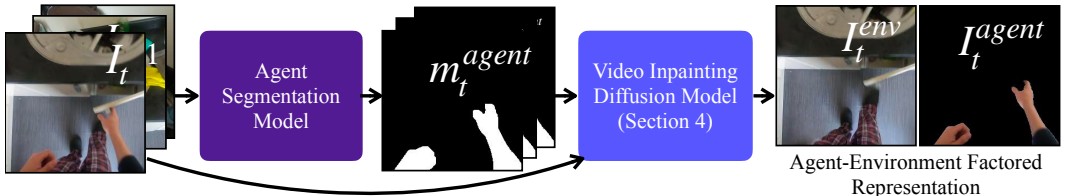

**Figure 1: Agent-Environment Factorization of Egocentric Videos.** Occlusion and the visual mismatch between humans and robots, make it difficult to use egocentric videos for robotic tasks. We propose a pixel-space factorization of egocentric videos into agent and environment representations (AEF, Sec. 3). An agent representation $I_t^{\text{agent}}$ is obtained using a model to segment out the agent. The environment representation $I_t^{\text{env}}$ is obtained by inpainting out the agent from the original image using VIDM, a novel Video Inpainting Diffusion Model (Sec. 4). AEF enables many different visual perception and robotics tasks (Fig. 2 and Sec. 5).

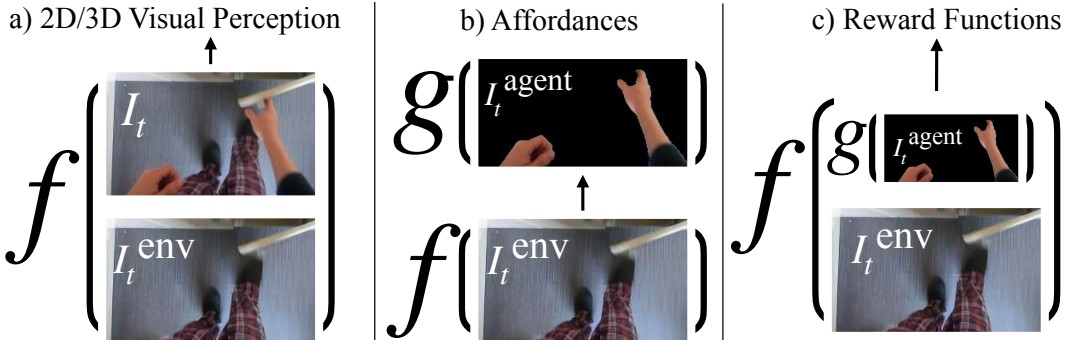

**Figure 2: Agent-Environment Factored (AEF) representations enable many applications.** (a) For visual perception tasks (Sec. 5.2 and 5.3), the unoccluded environment in $I_t^{\text{env}}$ can be used in addition to the original image. (b) For affordance learning tasks (Sec. 5.4), the unoccluded environment $I_t^{\text{env}}$ can be used to predict relevant desirable aspects of the agent in $I_t^{\text{agent}}$. (c) For reward learning tasks (Sec. 5.5 and 5.6) agent representations can be transformed into agent-agnostic formats for more effective transfer across embodiments.

But how do we obtain such a factored representation from raw egocentric videos? While detection & segmentation of hands are well-studied [13, 70, 103], our focus is on inpainting. Here, rather than just relying on a generic prior over images, we observe that the past frames may already have revealed the true appearance of the scene occluded by the hand in the current time-step. We develop a video inpainting model that leverages both these cues. We use a large-scale pre-trained diffusion model for the former and an attention-based lookup of information from the past frames for the latter. We refer to it as *Video Inpainting via Diffusion* (VIDM). Our approach outperforms DLFormer [62], the previous state-of-the-art for video inpainting, by a large margin (Tab. 1) and is $8\times$ faster at test time.

Next, we demonstrate the ability of the factored representation across tasks spanning 2D/3D visual perception to robot learning. Specifically, we adopt 5 existing benchmark tasks: a) 2D detection of objects of interaction [13], b) 3D reconstruction of hand-held objects [59, 95], c) learning affordances (where to interact and how) from egocentric videos [19], d) learning reward functions, and e) their use for interactive policy learning [1]. We show how selectively choosing and modifying aspects of the factored representation improves performance across all of these tasks compared to existing approaches. We believe the advances presented in this paper will enable the use of egocentric videos for learning policies for robots.

## 2    Related Work

**Robot Learning from Hand-Object Interaction Data.**  Many past works have sought to use human-object interaction data for robotic tasks. Researchers have used videos to predict: regions of interactions [19, 50], grasps / hand pose afforded by objects [19, 96], and post-grasp trajectories [2, 42].

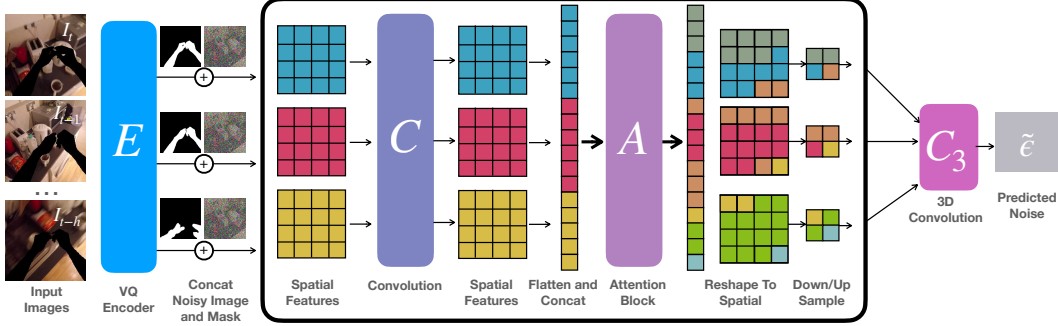

**Figure 3: Video Inpainting Diffusion Models (VIDM).** We extend pre-trained single-frame inpainting diffusion models [63] to videos. Features from context frames $(I_{t-h}, \ldots, I_{t-1})$ are introduced as additional inputs into the Attention Block $A$. We repeat the multi-frame attention block 8 times (4 to encode and 4 to decode) to construct the U-Net [66] that conducts 1 step of denoising. The U-Net operates in the VQ encoder latent space [63].

[59, 95] predict 3D shapes for hand-held objects. [1, 8, 69] learn reward functions from videos for learning real-world policies. Others use human-object interaction videos to pre-train feature representations for robotic policies [51, 60, 91]. While some papers ignore the gap between the hand and the robot end-effector [51, 60, 91], others adopt ways to be *agnostic* to the hand by masking it out [19], inpainting it [1], converting human pixels to robot pixels [73, 92], or learn embodiment invariant representations [8, 69, 100]. Instead, we pursue a factored representation that allows downstream applications to choose how to use the agent and the environment representations. Different from [103], we remove only the hand and retain the object of interaction.

**Diffusion Models** have been successful at unconditional [28, 74, 77] and conditional [15, 55, 63] image synthesis for several applications, *e.g.* text-to-image generation [3, 54, 61], image editing [27, 43, 48], video generation [21, 30, 86], video prediction and infilling [31] (similar to video inpainting, but focused on generating entire frames), 3D shape synthesis [45, 101, 106], self-driving applications [72, 107]. While diffusion models produce impressive results, they are computationally expensive due to the iterative nature of the generative process. To mitigate this limitation, several improvements have been proposed, *e.g.* distillation [44, 68], faster sampling [16, 75, 88], cascaded [29] and latent [63, 83] diffusion models. Among these, latent diffusion models (LDM) are the most commonly used variant. LDMs use an autoencoder to compress the high-dimensional image inputs to a lower-dimension latent space and train diffusion models in this latent space. In our work, we build off LDMs.

**Inpainting** requires reasoning about the local and global structure of the image to perform image synthesis. While earlier approaches focus on preserving local structure [4, 11, 17, 23], they are unable to capture complex geometries or large missing areas. Recent works alleviate these issues using neural networks [38, 40, 47, 58] to incorporate large receptive fields [46, 80], intermediate representations (*e.g.* edges [52, 93], segmentation maps [78]), adversarial learning [33, 78, 97, 99, 104]. With advancements in generative modeling, diffusion models have also been very effective on this task [43, 54, 63, 67, 96]. These approaches operate by learning expressive generative priors. An alternative is to utilize videos [36, 41, 62, 90] to reason about temporal consistency [32, 53, 87], leverage optical flow [79, 94], utilize neural radiance fields [81, 82], or extract context about the missing parts from the previous frames [35–37, 87]. Recent video-based inpainting methods incorporate transformers [41, 62, 102], cycle-consistency [90] & discrete latent space [62] to achieve state-of-the-art results. While existing approaches inpaint the entire video, we focus on inpainting a specific frame (previous frames are used as context) as required by downstream tasks. Our work adopts diffusion models for video-based inpainting to learn factored agent-environment representations, specifically, building on the single-frame inpainting model from [63].

## 3 Agent-Environment Factored (AEF) Representations

Motivated by the recent success of generative models, we develop our factorization directly in the pixel space. Given an image $I_t$ from an egocentric video, our factored representation decomposes

it into $I_t^{\text{agent}}$ and $I_t^{\text{env}}$. Here, $I_t^{\text{env}}$ shows the environment without the agent, while $I_t^{\text{agent}}$ shows the agent (Fig. 1) without the environment. These two images together constitute our Agent-Environment Factored (AEF) Representation.

**Using the Factorization.** This factorization enables the independent use of agent / environment information as applicable. For example, when hands are a source of nuisance (*e.g.* when object detectors are pre-trained on non-egocentric data without hands in Fig. 2a), we can use $I_t^{\text{env}}$ in addition to $I_t$ to get a better view of the scene. In other situations, it may be desirable to predict agent properties afforded by different parts of the environments (Fig. 2b). In such a situation, $I_t^{\text{env}}$ can be used to predict the necessary aspects of $I_t^{\text{agent}}$. In yet other situations, while the location of the agent provides useful information, its appearance may be a nuisance (*e.g.* when learning dense reward functions from egocentric videos for finetuning policies for robots in Fig. 2c). In such a situation, models could be trained on $I_t^{\text{env}}$ and $g(I_t^{\text{agent}})$, where the function $g$ generates the necessary abstractions of the agent image.

**Extracting the Factorization from Egocentric Videos.** For $I_t^{\text{agent}}$, we employ segmentation models to produce masks for the agent in $I_t$. For egocentric videos, we use the state-of-the-art VISOR models [13] to segment out the hand. When the depicted agent is a robot, we use DeepLabV3 [9] pre-trained on MS-COCO [39] and fine-tuned using manually annotated robot end-effector frames. Robot end-effectors have a distinctive appearance and limited variety, hence we can train a high-performing model with only a small amount of training data. We denote the agent segmentation in $I_t$ by $m_t^{\text{agent}}$.

Extracting $I_t^{\text{env}}$ is more challenging because of significant scene occlusion induced by the agent in a given frame. Naively extracting $I_t^{\text{env}}$ by just masking out the agent leaves artifacts in the image. Thus, we inpaint the image to obtain $I_t^{\text{env}}$. We condition this inpainting on the current frame $I_t$ and previous frames from the video, *i.e.* $\{I_{t-h}, \ldots, I_{t-1}\}$ since occluded parts of the scene in $I_t$ may actually be visible in earlier frames. This simplifies the inpainting model, which can *steal* pixels from earlier frames rather than only relying on a generic generative image prior. We denote the inpainting function as $p(I_t, m_t^{\text{agent}}, \{m_{t-h}^{\text{agent}}, \ldots, m_{t-1}^{\text{agent}}\}, \{I_{t-h}, \ldots, I_{t-1}\})$, and describe it next.

# 4 Video Inpainting via Diffusion Models (VIDM)

The inpainting function $p(I_t, m_t^{\text{agent}}, \{m_{t-h}^{\text{agent}}, \ldots, m_{t-1}^{\text{agent}}\}, \{I_{t-h}, \ldots, I_{t-1}\})$ inpaints the mask $m_t^{\text{agent}}$ in image $I_t$ using information in images $I_{t-h}$ through $I_t$. The function $p$ is realized through a neural model that uses attention [85] to extract information from the previous frames. We train this neural network using latent diffusion [63], which employs denoising diffusion [28] in the latent space obtained from a pre-trained vector quantization (VQ) encoder [18, 84].

**Model Architecture.** We follow prior work [63] and train a model $\epsilon_\theta(z_u, u, c)$, where $z_u$ is the noisy version of the diffusion target $z$ at diffusion timestep $u$, and $c$ is the conditioning (note we denote diffusion timesteps as $u$ to avoid conflict with $t$, which refers to video timesteps). For each diffusion timestep $u$, this model predicts the noise $\epsilon$ added to $z$. This prediction is done in the latent space of the VQ encoder, and the diffusion model is implemented using a U-Net architecture [66] with interleaving attention layers [85] to incorporate conditioning on past frames. The use of attention enables *learning* of where to steal pixels from, eliminating the need for explicit tracking. Furthermore, this allows us to condition on previous frames while reusing the weights from diffusion models pre-trained on large-scale image datasets.

We work with $256 \times 256$ images. The VQ encoder reduces the spatial dimension to $64 \times 64$, the resolution at which the denoising model is trained. The commonly used input for single-frame inpainting with a U-Net diffusion model is to simply concatenate the masked image, mask, and noisy target image along the channel dimension [63]. This gives an input shape of $(2d + 1, 64, 64)$ where $d$ is the dimension of the latent VQ codes. This U-Net model stacks blocks of convolutions, followed by self-attention within the spatial features. VIDM also concatenates the noisy target $z_u$ with conditioning $c$ (*i.e.* the context images, $I_{t-h}, \ldots, I_t$, and masks, $m_{t-h}^{\text{agent}}, \ldots, m_t^{\text{agent}}$) however, we stack context frames (*i.e.* $t - h$ through $t - 1$ along a new dimension. As such, our U-Net accepts a $(2d + 1, h + 1, 64, 64)$ tensor, where $h$ is the number of additional context frames. We perform convolutions on the $h + 1$ sets of spatial features in parallel but allow attention across all frames at attention blocks. Consequently, we can re-use the weights of an inpainting diffusion model trained on

**Table 1: In-painting evaluation** on held-out clips from Epic-Kitchens [12]. Use of strong generative priors and past frames allows our model to outperform past works that use only one or the other.

| Inpainting Method | PSNR↑ | SSIM↑ | FID↓ | Runtime ↓ |
|---|---|---|---|---|
| Latent Diffusion [63] | 28.29 | 0.931 | 27.29 | 12.5s / image |
| Latent Diffusion (fine-tuned) | 28.27 | 0.931 | 27.50 | 12.5s / image |
| DLFormer [62] | 26.98 | 0.922 | 51.74 | 106.4s / image |
| VIDM (Ours) | **32.26** | **0.956** | **10.37** | 13.6s / image |
| VIDM trained with hands visible | 31.81 | 0.953 | 11.04 | 13.6s / image |

single images. The final spatial features are combined using a $(h + 1) \times 1 \times 1$ convolution to form the final spatial prediction. Following prior work, the loss is $\mathbb{E}_{z,c,\epsilon \sim \mathcal{N}(0,1),u}\big[\|\epsilon - \epsilon_\theta(z_u, u, c)\|_1\big]$.

**Training Dataset.** The data for training the model is extracted from Epic-Kitchens [12] and a subset of Ego4D [20] (kitchen videos). Crucially, we don't have real ground truth (*i.e.* videos with and without human hands) for training this model. We synthetically generate training data by masking out hand-shaped regions from videos and asking the model to inpaint these regions. We use hand segment sequences from VISOR [13] as the pool of hand-shaped masks. In addition, we also generate synthetic masks using the scheme from [63, 104]. We use a 3-frame history (*i.e.* $h = 3$). Context images are drawn to be approximately 0.75s apart. We found it important to not have actual human hands as prediction targets for the diffusion model. Thus, we omit frames containing hands from Ego4D (using hand detector from [70]) and do not back-propagate loss on patches that overlap a hand in Epic-Kitchens (we use ground truth hand annotations on Epic-Kitchens frames from VISOR). We end up with 1.5 million training frames in total.

**Model Training.** We initialize our networks using pre-trained models. Specifically, we use the pre-trained VQ encoder-decoder from [63] which is kept fixed. The latent diffusion model is pre-trained for single-frame inpainting on the Places [105] dataset and is finetuned for our multi-frame inpainting task. We train with a batch size of 48 for 600k iterations on 8 A40 GPUs for 12 days. At inference time we use 200 denoising steps to generate images.

Our overall model realizes the two desiderata in the design of a video inpainting model. First, conditioning on previous frames allows occluded regions to be filled using the appearance from earlier frames directly (if available). Second, the use of a strong data-driven generative prior (by virtue of starting from a diffusion model pre-trained on a large dataset) allows speculation of the content of the masked regions from the surrounding context.

## 5 Experiments

We design experiments to test the inpainting abilities of VIDM (Sec. 5.1), and the utility of our factorized representation for different visual perception and robot learning tasks (Sec. 5.2 to Sec. 5.6). For the former, we assess the contribution of using a rich generative prior and conditioning on past frames. For the latter, we explore 5 benchmark tasks: object detection, 3D object reconstruction, affordance prediction, learning reward functions, and learning policies using learned rewards. We compare to alternate representations, specifically ones used in past papers for the respective tasks.

### 5.1 Reconstruction Quality Evaluation

We start by assessing the quality of our proposed inpainting model, VIDM. Following recent literature [36, 41, 62], we evaluate the PSNR, SSIM and FID scores of inpainted frames against ground truth images.

**Evaluation Dataset.** We select 33 video clips that do not contain any hands from the 7 held-out participants from the Epic-Kitchens dataset [12]. These clips are on average 3.5 seconds long and span a total of 5811 frames. For each clip, we mask out a sequence of hand-shaped masks mined from the VISOR dataset [13]. The prediction problem for inpainting models is to reproduce the regions underneath the hand-shaped masks. This procedure simulates occlusion patterns that models will need to inpaint at test time, while also providing access to ground truth pixels for evaluation.

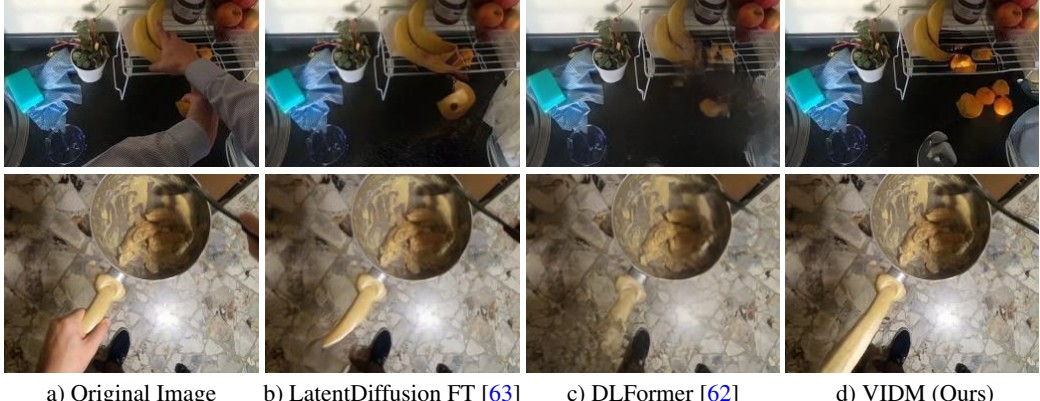

| a) Original Image | b) LatentDiffusion FT [63] | c) DLFormer [62] | d) VIDM (Ours) |

**Figure 4:** Our approach (VIDM) is able to correctly steal background information from past frames (top row, oranges on the bottom right) and also correctly reconstructs the wok handle using strong object appearance priors (bottom row).

**Table 2:** Average recall of detections from a COCO-trained Mask RCNN [24] on active objects (*i.e.* objects undergoing interaction) from VISOR [13]. See Sec. 5.2 for more details.

| Image Used | $AR_{all}@1$ | $AR_{all}@5$ | $AR_{all}@10$ | $AR_{0.5}@1$ | $AR_{0.5}@5$ | $AR_{0.5}@10$ |
|---|---|---|---|---|---|---|
| Raw Image (*i.e.* $I_t$) | 0.137 | 0.263 | 0.272 | 0.265 | 0.530 | 0.551 |
| Masked Image (*i.e.* $I_t$ with $m_t^{agent}$ blacked out) | 0.131 | 0.236 | 0.245 | 0.266 | 0.474 | 0.495 |
| $I_t^{env}$ inpainted using Latent Diffusion [63] | 0.149 | 0.259 | 0.270 | 0.299 | 0.517 | 0.541 |
| $I_t^{env}$ inpainted using Latent Diffusion (finetuned) | 0.154 | 0.262 | 0.271 | 0.305 | 0.519 | 0.540 |
| $I_t^{env}$ inpainted using VIDM (Ours w/o factorization) | 0.163 | 0.268 | 0.277 | 0.317 | 0.521 | 0.543 |
| $I_t$ and $I_t^{env}$ inpainted using VIDM (Ours w/ factorization) | **0.170** | **0.379** | **0.411** | **0.334** | **0.681** | **0.735** |

**Baselines.** We compare against 3 baseline models: a) the single-frame latent diffusion inpainting model [63] trained on the Places [105] dataset that we initialize from, b) this model but finetuned on *single images* from the same Epic-Kitchens dataset that we use for our training, c) *DLFormer* [62], the current state-of-the-art video inpainting model. DLFormer trains one model per clip and sees every frame in the test clip, unlike our model which only sees the previous 3 frames for each prediction. Finally, we include one ablation: VIDM trained with hands visible in the training objective (*i.e.* gradients propagate to all pixels) instead of the objective used in VIDM (stop-gradient on pixels with hands, see Section 3 for details).

**Results.** Tab. 1 reports the results. We can see our model outperforms all baselines. Processing context frames with attention allows our model to improve upon single-frame diffusion baselines. Training with multiple frames and hands visible improves performance, but this training procedure requires the model to output hands during training. Consequently, inpainted regions sometimes include hands at test time (see Figure S5). Our model also outperforms DLFormer [62], while being 8× faster and using only 3 frames of history. Fig. 4 shows example visualizations of inpainting results.

## 5.2 Application 1: Object Detection

The detection and recognition of objects being interacted with (by humans or robots) is made challenging due to occlusion caused by the hand / end-effector. While existing object detection datasets may include objects held and occluded by human hands, objects may be occluded in ways outside of the training distribution (*e.g.* when running COCO-trained detectors on egocentric videos). Furthermore, very few object detection datasets include examples of occlusion by a robot end-effector. We hypothesize that by removing the agent from the scene, off-the-shelf detectors may work better without requiring any additional training.

**Protocol.** We evaluate this hypothesis by testing COCO [39]-trained Mask RCNN detectors [24] on egocentric frames showing hand-object interaction from VISOR [13]. We only evaluate on object

**Table 3:** Average Precision on the Region-of-Interaction task and mAP for Grasps Afforded by Objects task from [19]. Using our model to remove the hands improves performance *vs.* placing a square mask over hands as done in [19]. See Sec. 5.4 for more details.

| Image Used | Region of Interaction (RoI) | | Grasps Afforded by Objects (GAO) |
|---|---|---|---|
| | 0% Slack | 1% Slack | |
| Masked image ($I_t$ with $m_t^{\text{agent}}$ blacked out) [19] | $47.50 \pm 2.77$ | $54.03 \pm 3.27$ | $35.53 \pm 3.67$ |
| $I_t^{\text{env}}$ (inpainted using VIDM) (Ours) | $\mathbf{50.77 \pm 0.81}$ | $\mathbf{57.10 \pm 1.01}$ | $\mathbf{41.00 \pm 3.99}$ |

classes that appear in both COCO and VISOR. VISOR [13] only annotates the active object and background objects have not been annotated. Thus, we measure performance using Average Recall (AR) as opposed to Average Precision. Specifically, we predict $k$ ($= 1, 5, 10$ in our experiments) boxes per class per image and report box-level Average Recall at $0.5$ box overlap ($\text{AR}_{0.5}$) and integrated over box overlap thresholds $\{0.5, 0.55, \ldots, 0.95\}$ ($\text{AR}_{\text{all}}$).

**Results.** Tab. 2 reports the average recall when using different images as input to the COCO-trained Mask RCNN detector. We compare against a) just using the raw image (*i.e.* $I_t$) as would typically be done, b) using a masked image (*i.e.* $I_t$ with $m_t^{\text{agent}}$ blacked out) a naive way to remove the hand, c) different methods for inpainting the agent pixels (pre-trained and finetuned single-frame Latent Diffusion Model [63] and our inpainting scheme, VIDM), and d) using both the original image and the inpainted image (*i.e.* both $I_t$ and $I_t^{\text{env}}$) as enabled by AEF. When using both images, we run the pre-trained detector on each image, merge the predictions, and return the top-$k$.

Naive masking introduces artifacts and hurts performance. Inpainting using a single-frame model helps but not consistently. Our video-based inpainter leads to more consistent gains over the raw image baseline, outperforming other inpainting methods. Our full formulation leads to the strongest performance achieving a 24%-51% relative improvement over the raw image baseline.

## 5.3 Application 2: 3D Reconstruction of Hand-Held Objects

Past work has tackled the problem of 3D reconstruction of hand-held objects [22, 34, 59, 95]. Here again, the human hand creates occlusion and hence nuisance, yet it provides valuable cues for the object shape [95]. Similar to Sec. 5.2, we hypothesize that the complete appearance of the object, behind the hand, may provide more signal to a 3D reconstruction model.

**Protocol.** We adopt the state-of-the-art approach from Ye *et al.* [95]. They design a custom neural network architecture that is trained with full supervision on the ObMan dataset [22]. Their model just accepts $I_t$ as input. To test whether the complete appearance of the object is helpful, we additionally input $I_t^{\text{env}}$ to their model. As ObMan is a synthetic dataset, we use *ground-truth $I_t^{\text{env}}$ images obtained by rendering the object without the hand.* This evaluation thus only measures the impact of the proposed factorization. We use the standard metrics: F1 score at 5 & 10 mm, and chamfer distance.

**Results.** Using $I_t$ and *ground truth* $I_t^{\text{env}}$ improves upon just using the raw image (as used in [95]) across all metrics: the F1 score at 5mm increases from 0.41 to 0.48, the F1 score at 10mm increases from 0.62 to 0.68, and the chamfer distance improves from 1.23 to 1.06. As we are using ground truth $I_t^{\text{env}}$ here, this is not surprising. But, it does show the effectiveness of our proposal of using factorized agent-environment representations.

## 5.4 Application 3: Affordance Prediction

Past works [19] have used videos to learn models for labeling images with regions of interaction and afforded grasps. While egocentric videos directly show both these aspects, learning is made challenging because a) the image already contains the answer (the location and grasp type exhibited by the hand), and b) the image patch for which the prediction needs to be made is occluded by the hand. Goyal *et al.* [19] mask out the hand and train models to predict the hand pixels from the surrounding context. This addresses the first problem but makes the second problem worse. After masking there is even more occlusion. This application can be tackled using our AEF framework, where we use inpainted images $I_t^{\text{env}}$ to predict aspects of the agent shown in $I_t^{\text{agent}}$.

**Table 4:** Spearman's rank correlations of reward functions learned on Epic-Kitchens [12] when evaluated on robotic gripper sequences for opening drawer, cupboard, and fridge tasks. Our factored representation achieves better performance than raw images or environment-only representation.

| Input Representation | Drawer | Cupboard | Fridge | Overall |
|---|---|---|---|---|
| Raw Imges (*i.e.* $I_t$) | 0.507 | 0.507 | 0.660 | 0.558 |
| Inpainted only (*i.e.* $I_t^{\text{env}}$ as proposed in [1]) | 0.574 | 0.570 | 0.671 | 0.605 |
| AEF ($I_t^{\text{env}}$ and $g(I_t^{\text{agent}})$) (Ours) | **0.585** | **0.582** | **0.676** | **0.614** |

**Protocol.** We adopt the training scheme from [19] (current state-of-the-art on this benchmark), but instead of using masked images as input to the model, we use images inpainted using our VIDM model. We adopt the same metrics and testing data as used in [19] and report performance on both the Region of Interaction (RoI) and Grasps Afforded by Objects (GAO) tasks. We test in a small data regime and only use 1000 images for training the affordance prediction models from [19]. We report mean and standard deviation across 3 trainings.

**Results.** Tab. 3 presents the metrics. Across both tasks, the use of our inpainted images leads to improved performance *vs.* using masked-out images.

## 5.5 Application 4: Learning Reward Functions from Videos

Difficulty in manually specifying reward functions for policy learning in the real world has motivated previous work to learn reward functions from human video demonstrations [1,6,9,71]. The visual mismatch between the human hand and robot end-effector is an issue. Past work [1] employs inpainting to circumvent this but consequently loses the important signal that the hand motion provides. Our factored representation in AEF provides a more complete solution and, as we will show, leads to improved performance over just using the raw data [9,71] and inpainting out the hand [1].

**Protocol.** We train a reward predictor on video clips from Epic-Kitchens [12] and assess the quality of the reward predictor on trajectories from a robotic end-effector. Specifically, we consider three tasks: opening a drawer, opening a cabinet, and opening a refrigerator. We use the action annotations from [12] to find video segments showing these tasks to obtain 348, 419, and 261 clips respectively. The reward function is trained to predict the task progress (0 at clip start and 1 at clip end) from a single image. The learned function is used to rank frames from trajectories of a robotic end-effector, specifically the reacher-grabber assistive tool from [76,98] captured using a head-mounted GoPro HERO7. We collected 10, 10, and 5 trajectories respectively for drawers, cupboards, and refrigerators. These robotic trajectories represent a large visual domain gap from the human-handed training demonstrations and test how well the learned reward functions generalize. Following [7], we measure Spearman's rank correlation between the rewards predicted by the learned reward function and the ground truth reward (using frame ordering) on the pseudo-robotic demonstrations. The Spearman's rank correlation only measures the relative ordering and ignores the absolute magnitude.

**Results.** Tab. 4 reports the Spearman's rank correlations for the different tasks. We compare the use of different representations for learning these reward functions: a) raw images (*i.e.* $I_t$), b) just inpainting (*i.e.* $I_t^{\text{env}}$), and c) our factored representation (*i.e.* $I_t^{\text{env}}$ and $I_t^{\text{agent}}$). For our factored representation, we map $I_t^{\text{agent}}$ through a function $g$ to an agent-agnostic representation by simply placing a green dot at the highest point on the end-effector mask (human hand or robot gripper) in the image plane (see Fig. 5), thus retaining information about where the hand is in relation to the scene. Our factored model, AEF, correlates the best with ground truth. Just painting out the end-effector loses important information about where the end-effector is in relation to objects in the scene. Raw frames maintain this information but suffer from the visual domain gap.

## 5.6 Application 5: Real-World Policy Learning using Learned Rewards

Motivated by the success of the offline evaluations in Sec. 5.5, we use the learned reward function for opening drawers to learn a drawer opening policy in the real world using a Stretch RE2 robot (see Fig. 5 (left)).

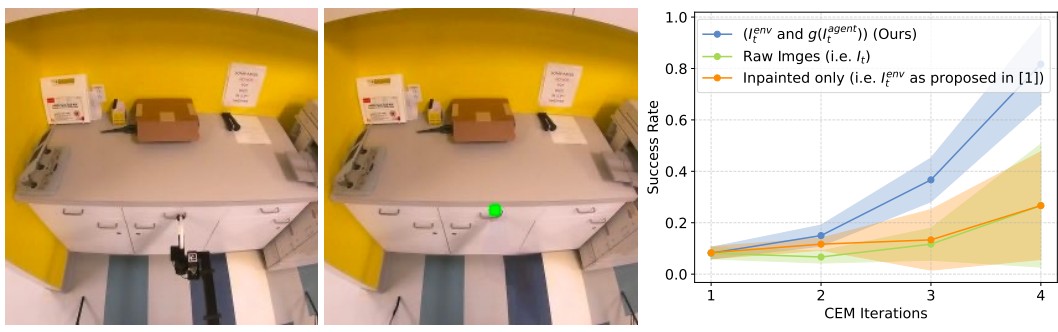

**Figure 5: Real-world experiment setup and results.** (left) Raw views from camera, (center) Inpainted image with agent-agnostic representation (green dot at top-pixel of end-effector). (right) Success rate as a function of CEM iterations.

**Protocol.** We position the robot base at a fixed distance from the target drawer. We use a 1D action space that specifies how far from the base the hand should extend to execute a grasp, before retracting. The drawer opens if the robot is able to correctly grasp the handle. We use a GoPro camera to capture RGB images for reward computation. The total reward for each trial is the sum of the predicted reward at the point of attempted grasp, and after the arm has been retracted. We use the Cross-Entropy Method (CEM) [14] to train the policy. We collect 20 trials in each CEM iteration and use the top-7 as *elite samples*. Actions for the next CEM iteration are sampled from a Gaussian fit over these elite samples. We use the success rate at the different CEM iterations as the metric.

**Results.** Similar to Sec. 5.5, we compare against a) raw images (*i.e.* $I_t$), b) inpainted images (*i.e.* just $I_t^{\text{env}}$ as per proposal in [1]), and c) our factored representation, AEF. As before, we use a dot at the end-effector location as the agent-agnostic representation of $I_t^{\text{agent}}$. We use our VIDM for inpainting images for all methods. We finetune a DeepLabV3 [9] semantic segmentation model (on 100 manually annotated images) to segment out the Stretch RE2 end-effector. Fig. 5 (right) shows the success rate as a function of CEM iterations. Reward functions learned using our AEF representation outperform baselines that just use the raw image or don't use the factorization.

## 6    Discussion

Our experiments have demonstrated the effectiveness of our inpainting model. Specifically, the use of strong priors from large-scale pre-trained inpainting diffusion models and the ability to steal content from previous frames allows VIDM to outperform past methods that only use one or the other cue (pre-trained LDM [63] and DLFormer [62] respectively). This is reflected in qualitative metrics in Tab. 1 and also in qualitative visualizations in Fig. 4.

Our powerful video inpainting model (along with semantic segmentation models that can segment out the agent) enables the extraction of our proposed Agent-Environment Factored (AEF) representation from egocentric video frames in pixel space. This allows intuitive and independent manipulation of the agent and environment representations, enabling a number of downstream visual perception and robotics applications. Specifically, we demonstrated results on 5 benchmarks spanning 3 categories. For visual perception tasks (object detection and 3D reconstruction of hand-held objects), we found additionally using the environment representation, which has the agent removed, improved performance. For affordance prediction tasks (region of interaction and grasps afforded by objects), we found that using the inpainted environment representation to be more effective at predicting relevant aspects of the agent representation than naive masking of the agent as used in past work. For robotic tasks (learning rewards from videos), the factored representation allowed easy conversion of the agent representation into an agent-agnostic form. This led to better transfer of reward functions across embodiments than practices adopted in past work (using the original agent representation and ignoring the agent altogether) both in offline evaluations and online learning on a physical platform.

Given that all information comes from the masked input frames, one might wonder, what additional value does the proposed inpainting add? First, it provides the practical advantage that each downstream application doesn't need to separately pay the implicit training cost for interpreting

masked-out images. Inpainted images are like real images, allowing ready re-use of the off-the-shelf models. Furthermore, the data complexity of learning a concept under heavy occlusion may be much higher than without. A *foundation* inpainting model can leverage pre-training on large-scale datasets, to inpaint the occlusion. This may be a more data-efficient way to learn concepts as our experiments have also shown.

## 7 Limitations and Broader Impact

As with most work with diffusion models, inference time for our method is 13.6 seconds which renders real-time use in robotic applications infeasible. AEF only inpaints out the hand to recover the unoccluded environment. There may be situations where the environment might occlude the agent and it may be useful to explore if similar inpainting could be done for agent occlusion.

At a broader level, while our use of the video inpainting model was for robotic applications, the VIDM model can also be thought of a general-purpose video inpainting model. This inherits the uncertain societal implications of generative AI models.

## Acknowledgments and Disclosure of Funding

We thank Ruisen Tu and Advait Patel for their help with collecting segmentation annotations for the claw and the robot end-effector. We also thank Arjun Gupta, Erin Zhang, Shaowei Liu, and Joseph Lubars for their feedback on the manuscript. This material is based upon work supported by NSF (IIS2007035), DARPA (Machine Common Sense program), NASA (80NSSC21K1030), an Amazon Research Award, an NVidia Academic Hardware Grant, and the NCSA Delta System (supported by NSF OCI 2005572 and the State of Illinois).

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

## S1    VIDM Training Details

For an overview of the VIDM model architecture, see Section 4. In each block, the CNN layers are implemented as residual blocks [25] with SiLU non-linearities [26] and each attention layer does self-attention across all token from all input images using 32-channel GroupNorm normalization. Following [63], upsampling and downsampling operations are both implemented using residual CNN blocks with either an internal nearest mode $2\times$ upsampling operation or internal $2\times$ downsampling via average pooling. An initial convolution brings the feature dimension to 256, which is raised to a maximum of 1024 at the center of the U-Net. At the highest spatial resolution of $64 \times 64$ the self-attention layer is omitted, as attention with 16384 ($= 64 \times 64 \times 4$) tokens is computationally intractable for our available hardware. The largest attention layer occurs at a spatial resolution of $32 \times 32$ across four images for a total of 4096 tokens.

We trained VIDM using target images from Ego4D [20] and VISOR [13] (see Section 4). Since no evaluation was done on Ego4D, no Ego4D data was held out. For VISOR, all data from participants

*P37, P35, P29, P05*, and *P07* was held-out from training. This held-out data from these participants was used for reconstruction quality evaluation (Section 5.1) and object detection (Section 5.2) experiments. Table S1 lists hyper-parameters. Figure S3 shows sample training batches.

**Table S1:** VIDM Model and Training Hyper-parameters.

| Hyper-parameter | Value |
|---|---:|
| Learning Rate | $4.8 \times 10^{-5}$ |
| Batch Size | 48 |
| Optimizer | Adam |
| Diffusion Steps (training) | 1000 |
| Latent image Size | $64 \times 64$ |
| Number of VQ Embedding Tokens | 8192 |
| VQ Embedding Dimension | 3 |
| Diffusion Steps (inference) | 200 |
| Attention Heads | 8 |

## S2 Downstream Task Experimental Details

### S2.1 Detection

We used off-the-shelf Mask R-CNN R_101_FPN_3x from Detectron2 [24, 89] trained on the COCO dataset [39] for evaluation. We used overlapping classes between the VISOR [13] annotations and COCO for evaluation. These were: *apple, banana, bottle, bowl, broccoli, cake, carrot, chair, cup, fork, knife, microwave, oven, pizza, refrigerator, sandwich, scissors, sink, spoon, toaster*.

### S2.2 Affordance Prediction

**Dataset:** We experiment on EPIC-ROI and GAO tasks from Goyal *et al.* [19]. EPIC-ROI uses the EPIC-KITCHENS dataset [12] and GAO uses YCB-Affordance [10] dataset. We consider a low data regime in our work and sample $1K$ images from these datasets to train the different models. For EPIC-ROI, we sample images with a probability inversely proportional to the length of the video. For GAO, we sample randomly. We use the same evaluation setting from [19].

**Model:** We use the same architecture from ACP [19] and replace the EPIC-ROI input images with images produced by our inpainting model (with hands removed) to incorporate our factorized representation. While ACP [19] masks out a patch at the bottom center of the image to hide the hand, we do not need any mask (neither for training nor for testing) since the hands have been removed via inpainting. The input is processed by ResNet-50 followed by different decoders for EPIC-ROI and GAO tasks.

**Training:** We train separate models for EPIC-ROI and GAO using the loss function and hyperparameters from ACP [19]. While it is possible to train a single model in multitask manner, we observe that the two tasks are not complementary to each other. We train using 3 seeds for each task and report the mean and standard deviation in the metrics.

### S2.3 3D Reconstruction of Hand-held Objects

**Dataset:** We use ObMan [22] dataset which consists of $2.5K$ synthetic objects from ShapeNet [5]. We use the train and test splits provided by Ye *et al.* [95]. We divide the train split into train and val set. The train set consists of $134K$, val set $7K$ and test set $6.2K$ images. The dataset provides 3D CAD models for each object, which we use for training hand-held object reconstruction model from Ye *et al.* [95].

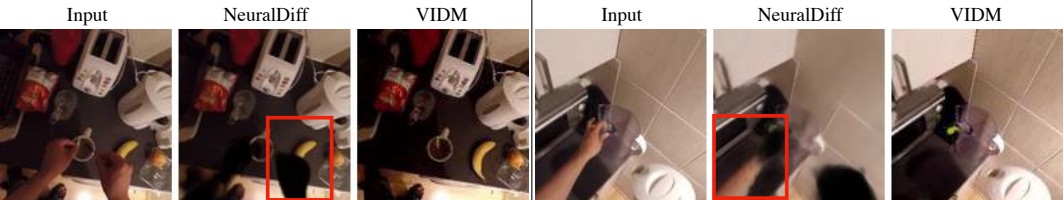

**Figure S1:** The environment factorization obtained by NeuralDiff contains artifacts, *e.g.* (left) region around banana, (right) partial hand. Our VIDM model produces more realistic results.

**Model:** We use the architecture from Ye *et al.* [95]. It uses FrankMocap [65] to extract hand articulation features from a single image using MANO [64] hand parameterization. These hand features are used as conditioning to a DeepSDF [56] model which predicts the object shape using implicit representation. This model also takes in pixel-aligned features and global image features along with hand features. To incorporate our factorized representation, we also extract global image features and pixel-aligned features from ObMan images showing only objects (with hands removed). These features are concatenated with the features from the input ObMan images and fed as input to the DeepSDF [56] decoder.

**Training:** Following [95], we use a normalized hand coordinate frame for sampling points and predicting SDFs. We sample 8192 points in $[-1, 1]^3$ for training, out of which half of them lie inside and the rest lie outside the object. At test time, $64^3$ points are sampled uniformly in $[-1, 1]^3$. We train the model in a supervised manner using 3D ground truth from ObMan [22] for 200 epochs with a learning rate of $1e - 5$. Other hyper-parameters are used directly from [95].

## S3  Additional Results

### S3.1  NERF Comparison

Neural radiance fields [49] represent an alternative path to producing agent-environment factorizations. We compare against NeuralDiff [82] as a representative method. We can only compare on the EPIC-Kitchens P05_01 sequence since it is the only one model released from that project that is common with our test set. We focus on frames that include a hand, and use their static and transient reconstruction as the prediction for $I^{\text{env}}_{\cdot}$ We contrast it with the prediction for from our model.

Figure S1 shows qualitative comparisons. On these images, our model achieves superior FID scores - 186.79 for VIDM vs 215.90 for NeuralDiff. Note that FIDs are overall higher than usual, but for good reason. There is no hand-removed image set (ie. objects floating in air) to use as reference to compute FID. As a proxy reference set, we use images that don't contain hands as reference and thus FID scores for both models are higher than usual.

### S3.2  Robot Learning: Pick-Up-Plate

To demonstrate that our method can work on tasks beyond opening, we followed the same protocol as in Table 4 for a fourth task of picking up a plate. In epic kitchens, there are less than $\frac{1}{3}$ as many sequences for this task as for opening drawers, and the quality is worse (clips having the plate out of frame, annotation timing being off etc.). This lack of data and low quality hurts generalization for all methods, but we still see a positive trend where using VIDM inpainted images with factorization gives Spearman's correlation of 0.139, while raw images and non-factorized inpainting give 0.118 and 0.083 respectively. We note that many other cross embodiment learning techniques may be used with our factored representation to explore more complex or multi-stage tasks (e.g [1, 7]) which we leave to future work.

### S3.3  Average Precision for Object Detection

In addition to the recall results reported in Table 2 we performed an small experiment to access our methods effect on object detection precision. To this end we took the class with the fewest

| $I_{t-3}$ | $I_{t-2}$ | $I_{t-1}$ | $I_t$ | $I_t$ in-painted (output) |

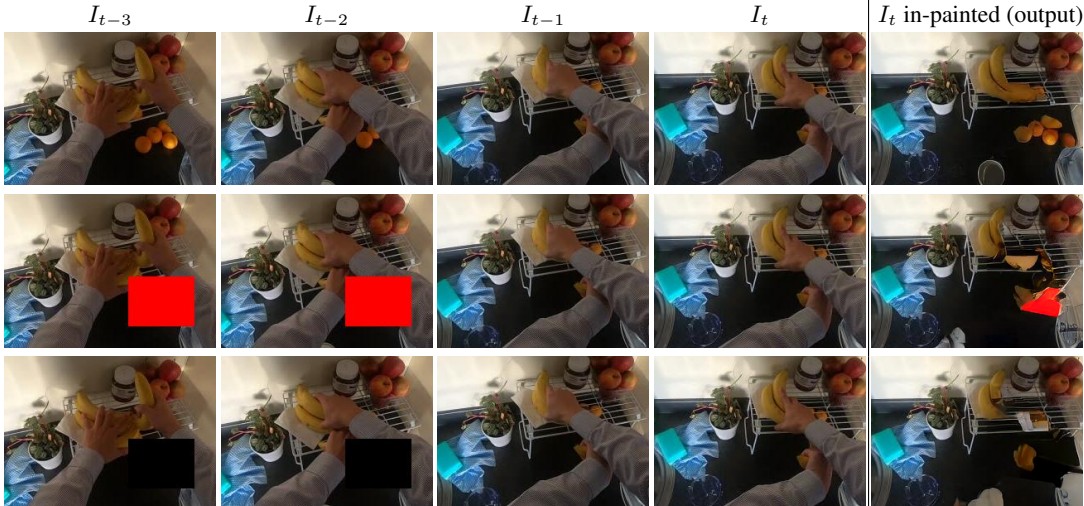

**Figure S2:** When the oranges are occluded by the hand, but visible in the previous frame, VIDM is able to reconstruct the oranges using this information (first row). When the oranges in previous frames are covered by either black or red masks, the model fails to reconstruct the oranges and inpaints red or black pixels instead (second and third rows). This suggests that the model does refer to content from the context frames for in-painting.

false positives (which happened to be 'scissors') when using raw images, and manually labeled all instances which were indeed true positives (adding missing detections to the ground truth labels). For this class with labels updated, using raw images only achieves an AP of 0.738, while using images in painted with VIDM achieves an AP of 0.762.

## S4 Visualizations

In Figure S3, we include a visualization of a training batch for our method, showcasing supervision and generated masks. In Figure S4, we include additional visualizations of the predictions made by our method and baselines.

In Figure S2 we visualize how VIDM responds to corruptions in the input images to highlight VIDM's ability to copy information from the context images. For this visualization we use a sequence of images where the region occluded by the hand is visible in the context frames. We manually corrupt the context frames by masking out (with both black and red boxes) the region from which the model needs to draw information to inpaint properly. We see that when presented with corrupted context frames, VIDM copies the corrupted pixels (black or red) to represent the occluded region, while it properly inpaints the region when using the uncorrupted frames.

Qualitative visualizations in Figure S5 exhibit the failure mode of training with hands visible. Because this training procedure requires the model to output hands some of the time, it sometimes paints the hand like pixels back into the image.

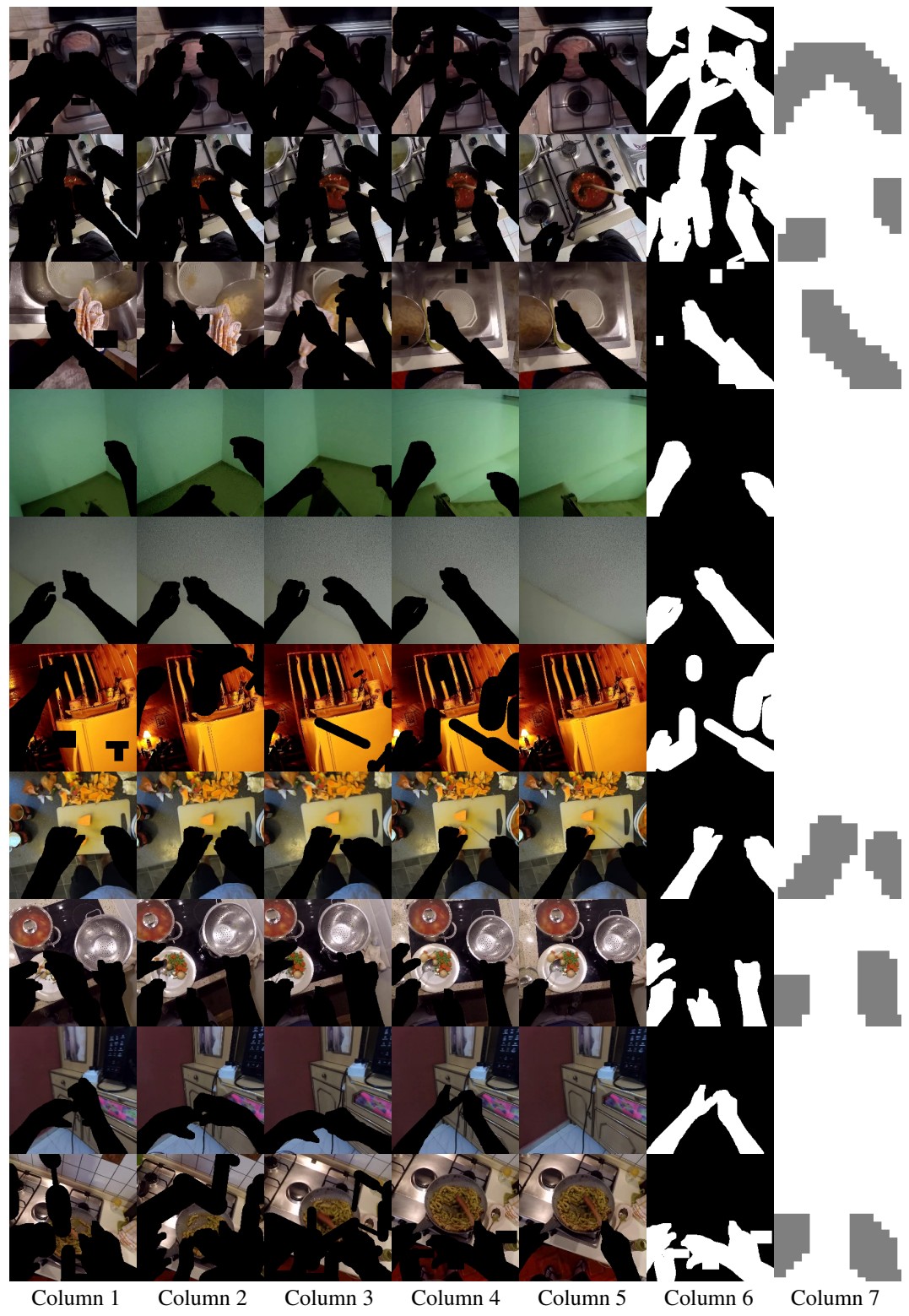

| Column 1 | Column 2 | Column 3 | Column 4 | Column 5 | Column 6 | Column 7 |

**Figure S3:** An example batch for training VIDM. Columns 1-4: Input images to the network. Column 5: target image for reconstruction. Column 6: Masked regions on the target image. Column 7: Pixels with loss propagated (white pixels have loss, gray pixels have no loss). Note that hands that are masked in the target image (column 5) have no loss on them. See Section 4 for details.

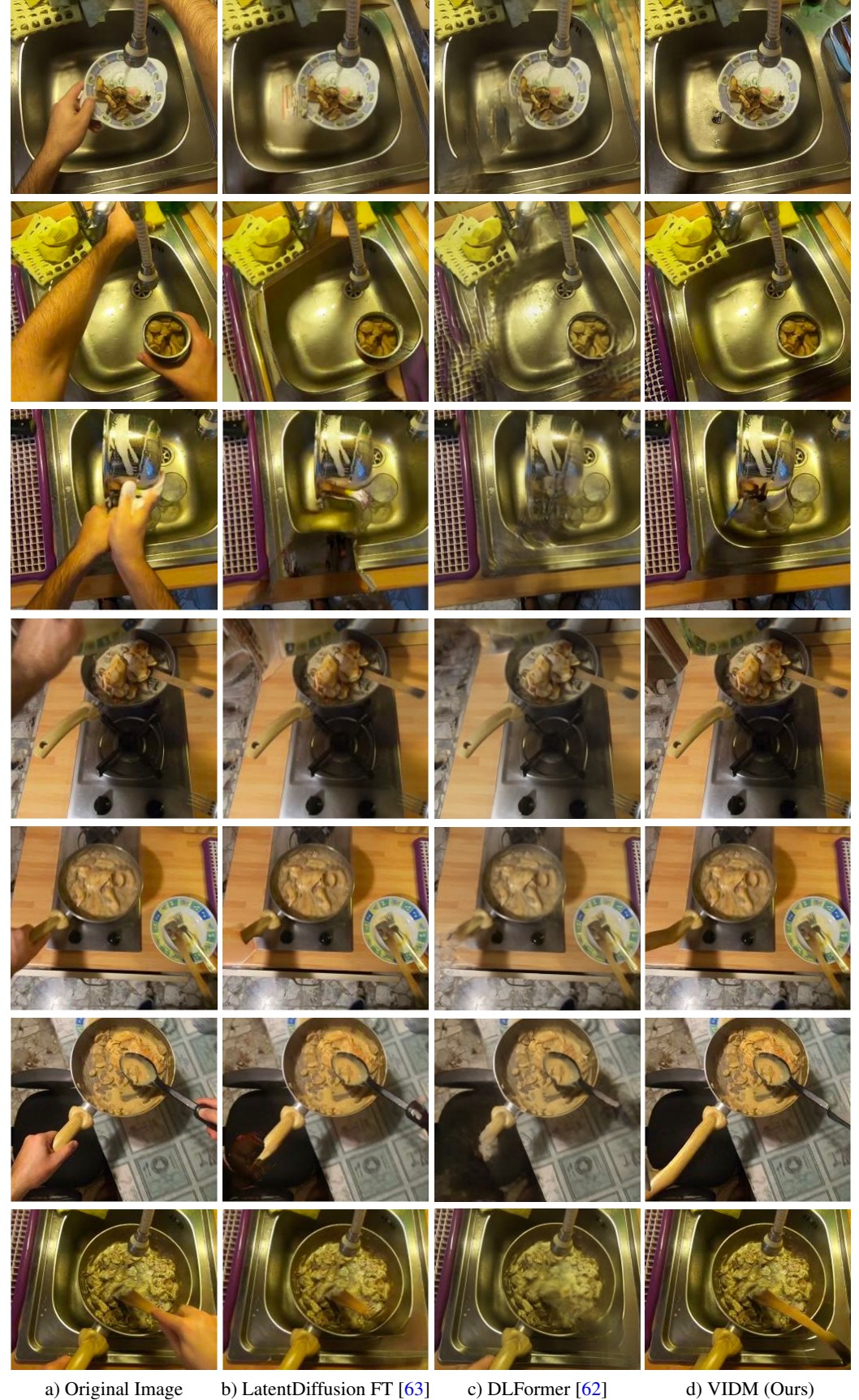

a) Original Image     b) LatentDiffusion FT [63]     c) DLFormer [62]     d) VIDM (Ours)

**Figure S4:** Additional visualizations of predictions from our method and baselines.

| Training /w hands | VIDM | Latent Diffusion |
| --- | --- | --- |

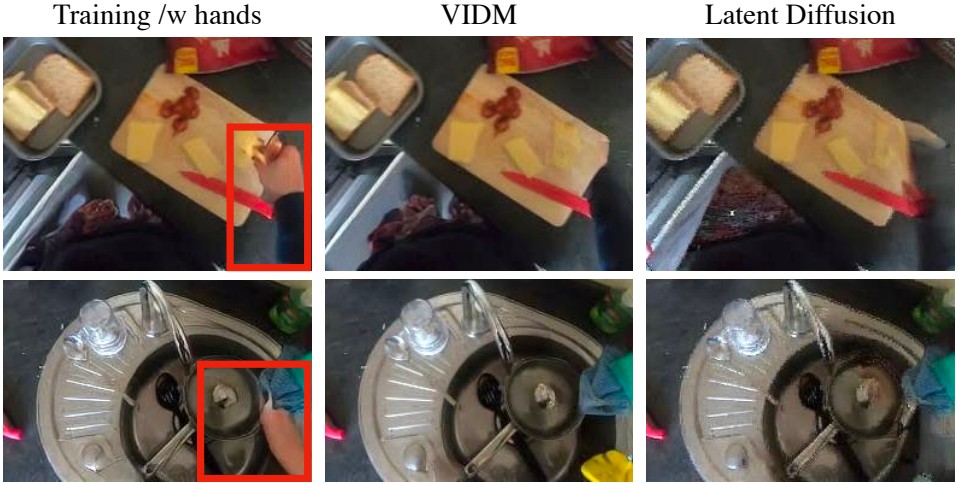

**Figure S5:** Training while propagating loss from pixels containing the hand, the model learns to reconstruct hands in occluded regions (left). This behavior does not occur with our training procedure, which does not apply loss on pixels with hands (center), and does not appear in base single-image model that has no egocentric specific finetuning (right).

