# OpenReview forum: "Look Ma, No Hands! Agent-Environment Factorization of Egocentric Videos"
_NeurIPS.cc/2023/Conference — NeurIPS 2023 poster_

### Official Review · Reviewer_t2iv · 2023-06-27

**Soundness:** 3 good
**Presentation:** 4 excellent
**Contribution:** 4 excellent
**Rating:** 7
**Confidence:** 4

**Summary:**

The paper addresses challenges in using egocentric videos for robotics tasks, specifically the issues of occlusion and visual mismatch between the human hand and a robot end effector. To address these problems, this work proposes a factored representation of the scene that separates the agent (human hand) from the environment. This factorization is achieved via the proposed Video Inpainting via Diffusion Model (VIDM). Experiments demonstrate the effectiveness of VIDM in improving the inpainting quality and highlights the power of the factored representation for various downstream robotics tasks.

**Strengths:**

1) Paper is well-written and well-structured; the descriptions of the experimental protocols for each of the applications was very helpful.

2) Experiments are very comprehensive; I was happy to see evaluation on both the inpainting quality achieved by the proposed model, as well as the utility of its associated representations for various downstream robotics tasks.



**Weaknesses:**

1) The main weakness that stands out to me is that the current set of downstream robotics applications are not as convincing as they need to be. Specifically, in many of the tasks, it seems the inpainted environment $I_{t}^{\text{env}}$ is the key component, and not the factorized representation in its entirety.



**Questions:**

1) It seems mostly $I_{t}^{\text{env}}$ is being used; where does the agent representation come into play? In application 3, I assume you're using $I_{t}^{\text{env}}$ to predict the GT hand pixels, so $I_{t}^{\text{agent}}$ is unused? And in applications 4 and 5, the function $g$ which abstracts $I_{t}^{\text{agent}}$ simply returns a green dot representing the position of the end effector/hand. This is a reasonable exploration of the concept, but are there more concrete use cases where $I_{t}^{\text{agent}}$ obtained from segmentation models (as described in L101-103) actually interplays equally with the inpainted $I_{t}^{\text{env}}$?

2) It seems like in application 2, what the experiment is actually proving is that ground truth environment information without the agent improves 3D reconstruction (unsurprising), and not "the effectiveness of our proposal of using factorized agent-environment representations" (L228-229). In other words, there's no connection at all between this experiment and the factorized representation that you learn from VIDM.

3) In applications 4 and 5, neither of these tasks lend themselves to needing this agent-environment factorized representation. In particular, I feel that in opening a drawer, cupboard or fridge, the occlusion by the hand/end-effector does not significantly hamper the accomplishing of the task. Even in Figure 5, I see that the (salient) occlusion by the green dot in the agent-agnostic representation is almost more extreme than the occlusion by the robotic end-effector in the raw image. What's the intuition here, or could it simply be the case that the baselines were not tuned sufficiently? I ask because in Table 4, the "inpainted only" performance is much better than the "raw image" performance, but in Figure 5 for the real-world experiments, "inpainted only" actually fails entirely compared to "raw image." Can the authors provide some explanation for this?

**Limitations:**

Limitations are addressed well in the discussion section of the paper.

---

> ### Author Rebuttal · Authors · 2023-08-10
>
> Thanks for your valuable comments and thoughtful feedback. Please see our response below and refer to the supporting figures in the rebuttal PDF.
>
> > It seems mostly Ienv is being used; where does the agent representation come into play? In application 3, I assume you're using Ienv to predict the GT hand pixels, so Iagent  is unused? And in applications 4 and 5, the function g which abstracts Iagent  simply returns a green dot representing the position of the end effector/hand. This is a reasonable exploration of the concept, but are there more concrete use cases where Iagent obtained from segmentation models (as described in L101-103) actually interplays equally with the inpainted env?
> * To clarify: when making predictions for application 3, $I_{agent}$ is not used as an input to the model, $f$. However in the parlance of figure 2, aspects of $I_{agent}$ (the pose of the hand) is used to provide the supervision to train $f$ to predict grasps afforded by objects (GAO task) in Table 3 and Section 5.4.
>
>    More generally, many past works re-target human hand pose to robot pose (eg [DexMV], [DexVIP], [Robotic Telekinesis] among others) for tele-operation and imitation learning. These applications involve using $I_{agent}$ to infer hand pose (and not just grasp type as in our work). One could also imagine a content creation application where $I_{agent}$ is used to predict aspects of $I_{env}$ (what object would fit into an animated character’s hand, for instance). Thus, we believe there are many other applications of our proposed factorization (and specifically $I_{agent}$)  than what we were able to experiment with in our paper.
>
> > It seems like in application 2, what the experiment is actually proving is that ground truth environment information without the agent improves 3D reconstruction (unsurprising), and not "the effectiveness of our proposal of using factorized agent-environment representations" (L228-229). In other words, there's no connection at all between this experiment and the factorized representation that you learn from VIDM.
> * Your understanding of application 2 is correct. However, the ground truth is exactly the ground truth version of $I_{env}$ in our proposed agent-environment factorization that our model tries to approximate. This experiment was designed to independently test the efficacy of AEF from VIDM, as such we use ground truth factorized images.
>
> > In applications 4 and 5, neither of these tasks lend themselves to needing this agent-environment factorized representation. In particular, I feel that in opening a drawer, cupboard or fridge, the occlusion by the hand/end-effector does not significantly hamper the accomplishing of the task. Even in Figure 5, I see that the (salient) occlusion by the green dot in the agent-agnostic representation is almost more extreme than the occlusion by the robotic end-effector in the raw image. What's the intuition here, or could it simply be the case that the baselines were not tuned sufficiently? I ask because in Table 4, the "inpainted only" performance is much better than the "raw image" performance, but in Figure 5 for the real-world experiments, "inpainted only" actually fails entirely compared to "raw image." Can the authors provide some explanation for this?
>
> * AEF offers a solution for both occlusion and the domain gap between human hands and the robot end-effector. In applications 4 and 5, the difficulty is not  so much about occlusion, but rather about the domain gap between human hands and the robot end effector. This domain gap is mitigated by the use of the same green dot across embodiments.
>
>    Excellent observation about Table 4 and Figure 5!  All methods learn to correctly score frames after the objects have been manipulated. But they have different behavior while the hand/robot end-effector is approaching the object. In Table 4, the camera motion in the egocentric claw videos provides a good signal for approaching the goal (thus high Spearman’s rho), while the robot experiment in Figure 5 has a fixed camera. The fixed camera in the robot experiment means that the reward function in the “Inpainted only” baseline doesn’t provide any feedback on the task progress, whereas our method, with its green dot, does. Hopefully, this clarifies your concern.
>
> We believe we have adequately addressed the concerns raised in this review. We will love to hear what you think and will be happy to offer further clarifications or respond to any other concerns. We hope our response helps improve the impression of our work.
>
> **References:**
>
> [DexMV] DexMV: Imitation Learning for Dexterous Manipulation from Human Videos. ECCV 2022.
>
> [DexVIP] DexVIP: Learning Dexterous Grasping with Human Hand Pose Priors from Video. CoRL 2021.
>
> [Robotic Telekinesis] Robotic Telekinesis: Learning a Robotic Hand Imitator by Watching Humans on YouTube. RSS 2022.

---

> > ### Comment · Reviewer_t2iv · 2023-08-15
> >
> > I thank the authors for their response; my concerns have been sufficiently addressed, and I have raised my rating to reflect this.

---

### Official Review · Reviewer_ymXH · 2023-07-03

**Soundness:** 3 good
**Presentation:** 3 good
**Contribution:** 2 fair
**Rating:** 5
**Confidence:** 4

**Summary:**

This paper proposes to use agent-environment factorization of egocentric videos to facilitate various downstream tasks (e.g., object detection, 3D reconstruction, affordance prediction, etc.). The authors leverages a pipeline to achieve agent-environment factorization. It consists of first a segmentation model that segment hands in egocentric videos, and second a diffusion-based inpainting model for filling the hand area. The benefit of the proposed pipeline is supported by improvements of baselins over various downstream tasks.

**Strengths:**

1. The idea of agent-environment factorization is interesting and as shown by the experiments, it does facilitate downstream tasks in various scenarios.
2. I do appreciate the sufficient amount of downstream tasks evaluated according, it proves the soundness of factorization.

**Weaknesses:**

1. One major concern on this paper is on its technical contribution. The proposed VIDM contains limited technical novelty as it is a basic segment-then-inpaint pipeline and does not propose any new modules.
2. There is no video-inpainting method compared in comparative experiments (e.g., [1]). As diffusion-based models usually perform poorly in inference speed, the current comparison with image-based inpainting models do not show whether VIDM's inference speed is enough for videos.
3. In the reward learning task (Sec. 5.5), the experiments are only conducted on 3 tasks, and all of them are open-action tasks. Would the results and analysis still hold for more complex tasks? or is it limited to the current selected domain.
4. In the real-world policy learning task (Sec. 5.6), the action space of the robot is very limited (1D) and has been placed in a very task-specific position. It might be too simple for making a point. Additionally, these experiments were only conducted on one task (still open-action) under one secneario, this makes it difficult to assess the generalizability and effectiveness.

**Questions:**

See the Weakness section. The authors could focus on:
1. Identify the uniqueness of the proposed VIDM and show its superiority for the current task instead of a plain pipeline.
2. Showing that agent-environment factorization could be beneficial for interaction tasks. As this could be the most important factor for the proposed pipeline, the current evaluated domain and task might be too limited.

**Limitations:**

The authors have adequately addressed the limitations.

---

> ### Author Rebuttal · Authors · 2023-08-10
>
> Thanks for your valuable comments and thoughtful feedback. Please see our response below and refer to the supporting figures in the rebuttal PDF.
>
> >One major concern on this paper is on its technical contribution. The proposed VIDM contains limited technical novelty as it is a basic segment-then-inpaint pipeline and does not propose any new modules.
> *  VIDM introduces new modules. It is more and better than just a segment-then-inpaint pipeline. Furthermore, our paper is more than just VIDM.
>
>    Specifically, VIDM uses cross-frame attention layers to transform a pre-trained image diffusion model into a video in-painter as described in Section 4 and Figure 3. We evaluate the effectiveness of this proposed architectural contribution on the video-inpainting task and observe improvements over a basic segment-then-inpaint pipeline as well as the current state-of-the-art for video inpainting. Furthermore, a major component of the paper is not just the performance of VIDM, but how we use VIDM. We propose a novel agent-environment factored representation for egocentric videos and show its effectiveness in extensive experimental evaluation across 5 benchmarks spanning 2D/3D perception to robot learning.
>
> >There is no video-inpainting method compared in comparative experiments (e.g., [1]). As diffusion-based models usually perform poorly in inference speed, the current comparison with image-based inpainting models do not show whether VIDM's inference speed is enough for videos.
>
> * Our paper already included comparisons to DLFormer, the state-of-the-art for video in-painting tasks in Table 1 and Section 5.1. We note a large improvement in metrics over this prior state-of-the-art (PSNR for DLFormer 26.98, vs 32.26 for ours). Table 1 in the DLFormer paper already reports comparison to the cut-and-paste in-painter used in [1] and reports very large improvements over cut-and-paste, thus we didn’t include a direct comparison to cut-and-paste.
>
>   Furthermore, we also report the inference speed in Table 1. VIDM needs 13.6s / image and is also much faster than 106.4s/image for DLFormer. This is not real-time but our applications don’t require real-time inference.
>
> > In the reward learning task (Sec. 5.5), the experiments are only conducted on 3 tasks, and all of them are open-action tasks. Would the results and analysis still hold for more complex tasks? or is it limited to the current selected domain.
>
> * To demonstrate that our method can work on tasks beyond opening, we followed the same protocol as in Table 4 for a fourth task of picking up a plate. In epic kitchens, there are less than ⅓ as many sequences for this task as for opening drawers, and the quality is worse (clips having the plate out of frame, annotation timing being off etc.). This lack of data and quality hurts generalization for all methods, but we still see a positive trend where using VIDM inpainted images with factorization gives Spearman’s correlation of 0.139, while raw images and non-factorized inpainting give 0.118 and 0.083 respectively. We note that many other cross embodiment learning techniques may be used with our factored representation to explore more complex or multi-stage tasks (e.g [1,7]) which we leave to future work.
>
> > In the real-world policy learning task (Sec. 5.6), the action space of the robot is very limited (1D) and has been placed in a very task-specific position. It might be too simple for making a point. Additionally, these experiments were only conducted on one task (still open-action) under one secneario, this makes it difficult to assess the generalizability and effectiveness.
> *  While this experiment may be simple, it still does make a point. Past work in this setting [1] (which is represented by the orange line in figure 5 right) doesn’t work because information of where the end-effector is relative to the object of interaction is lost. This slows down learning. AEF factorization retains the end-effector position while minimizing the domain gap and consequently learns faster. Lack of cheap dexterous manipulators limits the tasks we can tackle in the real world.
>
>    In addition to improvements over baselines in 4 other applications (2d object detection, 3d shape prediction, affordance prediction, and offline reward learning) this real world experiment demonstrates real-world feasibility of AEF+VIDM.
>
> We believe we have adequately addressed the concerns raised in this review. We will love to hear what you think and will be happy to offer further clarifications or respond to any other concerns. We hope our response helps improve the impression of our work.

---

> > ### Comment · Reviewer_ymXH · 2023-08-22
> > **Post-rebuttal response**
> >
> > Thanks the authors for the clarification, the rebuttal has addressed most of my concerns, therefore I'm willing to increase my original rating to 5.

---

### Official Review · Reviewer_jD5i · 2023-07-06

**Soundness:** 3 good
**Presentation:** 3 good
**Contribution:** 3 good
**Rating:** 6
**Confidence:** 3

**Summary:**

This work proposes the use of a factored agent and environment representation to handle two ego-centric video problems introduced by human hands : 1. They occlude objects of interaction and induce a domain gap between the data available for learning (egocentric videos) and the data seen by the robot at execution time; 2. Removing hands from the scene by masking or inpainting abandon the information of object affordances. The paper demonstrates the ability of the factored representation across tasks spanning 2D/3D visual perception to robot learning. They also show how selectively choosing and modifying aspects of the factored representation improves performance across all of these tasks compared to existing approaches.

**Strengths:**

Strength
1. The paper is clearly written and easy to follow. The related work provides enough information for the reviewers to get familiar with the background of ego-centric video and related tasks.
2. The proposed diffusion model, VIDM, is effective while efficient. The performance looks amazing compared to previous work.
3. The experiment part is sound and shows the effectiveness of the proposed VIDM across many benchmarks.
4. The motivation is intuitive. Combining hand pose as well as inpainting technique can provide more information than before, and thus the improvement is plausible.

**Weaknesses:**

Weakness
1. In figure 2, what is the meaning of the big f and g? Does they stand for different functions? If so, what is the purpose of drawing them in that way? The idea of this figure is not that clear.
2. Which part in VIDM contributes the most to the performance improvement across those benchmark? There seems to be no related ablation study on this.
3. In table 1, how does the stable diffusion (fine-tuned) done? It would be great if the author could provide more detail on this.

**Questions:**

Please see my comment in weakness.

**Limitations:**

Please see my comment in weakness.

---

> ### Author Rebuttal · Authors · 2023-08-10
>
> Thanks for your valuable comments and thoughtful feedback. Please see our response below and refer to the supporting figures in the rebuttal PDF.
>
> > In figure 2, what is the meaning of the big f and g? Does they stand for different functions? If so, what is the purpose of drawing them in that way? The idea of this figure is not that clear.
> * Figure 2 is meant to illustrate a few of the ways one could use an agent-envorinment factorization. For example, in section a (left) we show $I_{env}$ and $I_{agent}$ being passed into a single function that optimizes some perception task. This mirrors application 1, which uses the both elements of the factorization to improve object detection. Similarly in section c (right) we show that one can independently transform the agent representation (with some function $g$) before passing into both elements into another function ($f$). This is particularly useful when working with multiple embodiments. We demonstrate this in applications 4 and 5, where we process $I_{agent}$ with a function ($g$ in the language of figure 2) that maps both robot end-effectors and human hands to the same visual representation (green dot). This processing remove the visual domain gap, improving generalization across embodiments. We will update the caption in this figure to be clearer.
>
> > Which part in VIDM contributes the most to the performance improvement across those benchmark? There seems to be no related ablation study on this.
> * In the paper we have already included an ablation, and we have added another for the rebuttal along with a diagnostic visualization as described below.
>
>   * In the paper we compare VIDM, vs Latent Diffusion finetuned on our data. This shows that there is a clear benefit to the multi-frame nature of our model, and gains aren’t just due to training on in-domain data.
>
>    * As for ablations on the nature of our training: we are currently running an ablation on the hand exclusion aspect of our training protocol (filtering out Ego4d frames with hands and not propagating loss on pixels with hands). Unfortunately it won’t finish in the time duration of the rebuttal (the model takes about 12 days to train). Preliminary quantitative results at 3 days of training indicate that this choice is indeed effective across all metrics (PSNR of 31.14 vs 32.17 for our original model at 3 days of training. SSIM: 0.950 vs 0.955, FID 12.10 vs 10.57). However, qualitative visualizations exhibit the error mode that we saw during our development. Because this ablated model has to output hands some of the time, it sometimes paints the hand like pixels back into the image. See examples in the PDF (Figure B2) attached with the main response.
>
>     *  Furthermore, in order to give some insight to how our model uses information from prior frames, we visualized how our method responds to corruptions in context frames at test time. See examples in the PDF (Figure B1) attached with the main response.
>
> > In table 1, how does the stable diffusion (fine-tuned) done? It would be great if the author could provide more detail on this.
> * We took the single frame model that we extended to multiple frames (Latent Diffusion inpainting pre-trained on Places) and simply finetuned it on the exact same data that we used for finetuning VIDM. Since the single frame model takes in no context frames it ignored the extra frames and was finetuned to inpaint the masked region in the target frame. The same hand exclusion techniques and other training choices were used for this experiment.
>
> We believe we have adequately addressed the concerns raised in this review. We will love to hear what you think and will be happy to offer further clarifications or respond to any other concerns. We hope our response helps improve the impression of our work.

---

> > ### Comment · Reviewer_jD5i · 2023-08-17
> >
> > The rebuttal solves my concern well, and I wish to keep my rating.

---

### Official Review · Reviewer_u9dg · 2023-07-06

**Soundness:** 3 good
**Presentation:** 4 excellent
**Contribution:** 3 good
**Rating:** 7
**Confidence:** 4

**Summary:**

The following work presents a factorized approach for video-based egocentric tasks. Specifically, they propose to break down the video feed into separate environment-only and hands only feeds. Intuition behind this formulation is that change in appearance of the hands may constitute a domain gap when the source of the video perspective changes (e.g. person to person, person to robot manipulator). Furthermore, explicit factorization provides the model with additional supervision on the breakdown between what is the environment and what is the manipulator.

The hands are removed from the video feed using a video-inpainting model based on the latent diffusion architecture with attention-based extensions to attend to multiple past frames. Results demonstrate that their video-inpainting formulation outperforms DLFormer in both the video inpainting task (limited to their use case) as well as in improvements to downstream applications (object detection, affordance prediction, etc.)

**Strengths:**

- Simple idea with extensive demonstration of improvements in multiple downstream tasks.
- Strong results for video inpainting in the ego-centric setting with similarly very simple but pretty easy to justify design choices (optical-flow-like attention for past frames, exclusion of hands from training data)

**Weaknesses:**

- While the superiority of their video inpainting formulation is demonstrated only within the hand-removal task of ego-centric videos, the language used to describe the method can often be misinterpreted as a broader claim for outperforming existing  state of the art models in a general sense.
- The authors found it helpful to exclude hands as prediction targets during training. This seems like a significant design decision that should have a corresponding ablation study.
- Additional training details for comparison against DLFormer are missing:
  - Was DLFormer also trained with the hand-exclusion technique?
  - On what training data were the visual codebooks used by the LDM formulation and DLFormer derived from?
- While I don't necessarily doubt the idea that the factorized formulation improves object detection, I do not see average recall as an appropriate replacement for average precision. I would much rather see average precision measured on a limited set of categories where all instances of the object category in question are annotated.

**Questions:**

See weaknesses

**Limitations:**

Technical limitations discussed. I'm not sure it's sufficient to simply say that this work inherits uncertain societal implications from other generative modeling works.

---

> ### Author Rebuttal · Authors · 2023-08-10
>
> Thanks for your valuable comments and thoughtful feedback. Please see our response below and refer to the supporting figures in the rebuttal PDF.
> > While the superiority of their video inpainting formulation is demonstrated only within the hand-removal task of ego-centric videos, the language used to describe the method can often be misinterpreted as a broader claim for outperforming existing state of the art models in a general sense.
>
> * Thanks for the feedback. We will take a pass and further qualify our claims to be more about the hand inpainting task in egocentric videos.
>
> > The authors found it helpful to exclude hands as prediction targets during training. This seems like a significant design decision that should have a corresponding ablation study.
>
> * We are running this ablation but, unfortunately it won’t finish in the time duration of the rebuttal (the model takes about 12 days to train). Preliminary quantitative results at 3 days of training indicate that this choice is indeed effective across all metrics (PSNR of 31.14 vs 32.17 for our original model at 3 days of training. SSIM: 0.950 vs 0.955, FID 12.10 vs 10.57). However, qualitative visualizations exhibit the error mode that we saw during our development. Because this ablated model has to output hands some of the time, it sometimes paints the hand like pixels back into the image. See examples in the PDF (Figure B2) attached with the main response.
>
> > Additional training details for comparison against DLFormer are missing: Was DLFormer also trained with the hand-exclusion technique?
>
> * Yes. DLFormer is a per-clip method. It fits a unique set of model weights for each clip at test-time. Since all hands are masked out during test-time inpainting, DLFormer never sees any hands during test-time finetuning. It uses no additional pre-training step beyond having a pre-trained visual codebook. This codebook can easily reconstruct images from EPIC without hands.
>
> > On what training data were the visual codebooks used by the LDM formulation and DLFormer derived from?
>
> * Because we are finetuning pre-trained models we had to use the same codebooks that were used for the released LDM and DLFormer models: Places for LDM and COCO for DLFormer. We verified that both codebooks did a comparable job at reconstructing frames from the EPIC Dataset.
>
>
> > While I don't necessarily doubt the idea that the factorized formulation improves object detection, I do not see average recall as an appropriate replacement for average precision. I would much rather see average precision measured on a limited set of categories where all instances of the object category in question are annotated.
>
> * This is a good experiment to run. To this end we took the class with the fewest false positives (which happened to be ‘scissors’) when using raw images, and manually labeled all instances which were indeed true positives (adding missing detections to the ground truth labels). For this class with labels updated, using raw images only achieves an AP of 0.559, while using images in painted with VIDM achieves an AP of 0.584.
>
> We believe we have adequately addressed the concerns raised in this review. We will love to hear what you think and will be happy to offer further clarifications or respond to any other concerns. We hope our response helps improve the impression of our work.

---

> > ### Comment · Reviewer_u9dg · 2023-08-19
> > **Concerns addressed**
> >
> > The authors did a great job of addressing all my concerns, as well as the concerns of many other reviews. I am increasing my rating to accept with the expectation that all changes are appropriately incorporated into the final draft.

---

### Official Review · Reviewer_UX1K · 2023-07-06

**Soundness:** 3 good
**Presentation:** 3 good
**Contribution:** 2 fair
**Rating:** 5
**Confidence:** 4

**Summary:**

The paper proposes agent-environment factorization (AEF) as a representation for egocentric videos. AEF consists of 2 parts: the hand segmentation as agent part, and video inpainted environment part. The former uses an off-the-shelf hand segmentation while the latter is from finetuning an inpainting diffusion model. The authors show several downstream applications to demonstrate the AEF can improve recognition, reconstruction, and robotic tasks.


**Strengths:**

+ The key idea is agent-environment factorization, which is shown beneficial to multiple applications, 2D recognition, 3D reconstruction, and robotic tasks. The baselines are carefully designed such that they are directly comparable and showcase where and why AEF helps.
+ In the design of video inpainting model, they use the pretrained image inpainting to get the spatial prior while using nearby frames to aggregate more context.


**Weaknesses:**

1. While the factorization is shown effective, the improvement over the video inpainting model itself appears a bit incremental. This is based on 1) qualitative results in FigS3 2) some domain specific knowledge being used (e.g. no loss on the hand pixels, copy paste hand-shaped masks, etc). It is not clear how well the proposed architecture can generalize to other well-adopted video inpainting benchmark benchmarks like DAVIS or Youtube VOS.

    1.1 The proposed architectures may not be as critical as the paper claims (like in Table 1, see 1.2). It seems finetuning an generative video model, e.g. a MAGVit or video diffusion model may lead to results as good as the current method shows. I wonder if the authors agree with my conjecture.

    1.2 The numbers in Table 1 indicate a significant gap over current SoTA video inpainting methods. Improvement over Latent Diffusion is sensible since the model sees more context. But DLFormer is a completely different method – a per-clip model that operates in pixel space.  All of the metrics would favor sharp high-frequency signals, which latent space in latent diffusions are good at but DLFormer lacks.

    1.3 visualizing attention may be helpful to understand how context in nearby frames help inpainting.

2. The paper is very related to “Neural Feature Fusion Fields” which factorize the videos into background, agent, and, in addition, moving objects. Although they optimize a per-clip representation, the differences with this line of works should be discussed.

3. There are some improvements across multiple downstreaming applications but the improvements are not surprising. For example, seeing the unoccluded objects / environment improves 3D reconstruction; merging prediction of both unoccluded and original images boosts object detection; seeing the hand location indicates frame order better.
    3.1 It is a minor point but in Table 2, it will be more fair to compare if the proposed region is also doubled for the baselines since the proposed method is evaluated with twice the predictions.



**Questions:**

Overall I think it is a sound paper. The novelty are faire and mainly empirical -- showing this factorization can help several downstream tasks. See weakness. My main concern is 1.1 and 2.

**Limitations:**

The author discussed the limitation explicitly.

---

> ### Author Rebuttal · Authors · 2023-08-10
>
> Thanks for your valuable comments and thoughtful feedback. Please see our response below and refer to the supporting figures in the rebuttal PDF.
>
> **Clarifications about improvement in the video inpainting model**
> First, to clarify, VIDM includes a) an architectural modification on top of an image-based diffusion model to use information from past frames via cross-frame attention, and b) domain specific insights to train VIDM for the application we are interested in (agent-environment factorization). In our view, improvements over the baselines are quite salient, both qualitatively and quantitatively. Our output in Figure S3 suffers from fewer artifacts (row 1, row 2, row 5, row 6) and better completes the objects (row 3, row 4, row 7). Quantitative improvements in Table 1 are also quite solid, PSNR improves from 28.27 to 32.26, FID decreases from 27.50 to 10.37. Note that baselines were retrained on the same dataset that we use, so this represents a solid improvement.
>
> In the setting that we care about (egocentric videos), the major challenges are large camera motion, and dynamic occlusion (hands holding objects and moving through the scene). In this setting we are able to outperform the current state-of-the-art (DLFormer). Since our downstream applications involve hand occlusion, we focus on the task of inpainting hands in egocentric video. For this use case, our model outperforms existing models and lets us build effective agent-environment factorizations.
>
> **Discussion about finetuning a video generative model**
> The reviewer’s suggested approach is feasible, but it is not trivial to apply in our setting. For starters, published video diffusion models at the time of submission  [19, 26, 75], could not be directly applied to our video inpainting task (they either operate at lower resolution or do not incorporate masks for inpainting). The reviewer’s suggested approach (taking a video prediction model, and adapting it to do inpainting), is similar to the approach we took (taking an inpainting model and adapting it to video). We started from an image-based generative model and made necessary modifications to get it to work for video in-painting. We had our reasons: LDM released code, code for MAGVit wasn’t available, other video generative models weren’t applicable for reasons above. So, yes we agree with your conjecture, and such alternatives are interesting avenues for future work. But that doesn’t nullify our contributions in this paper, that is AEF combined with VIDM.
>
>
> **Comparison with SoTA video inpainting methods.**
> Independent of how DLFormer works, we compared against DLFormer as it is, to the best of our knowledge, the state-of-the-art at the video in-painting task. PSNR and SSIM are standard metrics. DLFormer itself uses these metrics for which it reports state-of-the-art results. We also report FID which is not directly comparing any two images, but rather comparing image statistics computed across all inpainted test clips versus ground truth. We outperform the current state-of-the-art for video inpainting in our setting (hand removal in egocentric videos) and our contribution (video inpainting extension of latent diffusion models) leads to improvements over just image inpainting with latent diffusion.
>
> **Visualization to understand how context in nearby frames help inpainting.**
> Excellent suggestion! Figure B1 in the main response PDF, visualizes how our method responds to corruptions in context frames at test time. This suggests VIDM does use info from context frames when necessary..
>
>
> **Discussion about Neural Feature Fusion Fields and NeuralDiff.**
> Thanks for this pointer. This is a relevant reference that we will cite and discuss. Neural Feature Fusion Fields (like NeuralDiff) embeds inductive bias into NeRF to obtain a decomposition into a static background, transient foreground object, and agent. Using NeRF lets them infer a 3D factorization but it comes with its own limitations: a) it requires many (100s of) viewpoints to work, and b) there are no priors that can be used to complete objects that are never observed. In contrast, our method pursues a factorization in 2D, can inpaint reasonaly with just 4 frames of context, and can also use priors on appearance of objects from large-scale pre-trained diffusion models.
>
>
> We compare to NeuralDIff (NFFF didn’t release models for EPIC videos) on P05_01 sequence since it is the only one that is common with our test set. We focus on frames that include a hand, and use their static and transient reconstruction as the prediction for $I_{env}$. We contrast it with the prediction for $I_{env}$ from our model. Figure B3 shows qualitative comparisons. On these images, our model achieves superior FID scores - 186.79 for VIDM vs 215.90 for NeuralDiff. Note that FIDs are overall higher than usual, but for good reason. There is no hand-removed image set (ie. objects floating in air) to use as reference to compute FID. As a proxy reference set, we use images that don’t contain hands as reference and thus FID scores for both models are higher than usual.
>
> **Clarification about experiments in Table 2.**
> First, to clarify, all methods in Table 2 return the same number of proposals. The last row, that runs the detector twice, pools together the detections from the two runs but returns the same number of detections as the baselines. Thus, in our view, the comparison, as is, is fair. One could argue that we use 2x compute time than baselines. For this, the second to last row presents a direct comparison where we run the detector just once (but on $I_{env}$) and still see improvements over the raw image and other baselines across most metrics.
>
> We believe we have adequately addressed the concerns raised in this review. We will love to hear what you think and will be happy to offer further clarifications or respond to any other concerns. We hope our response helps improve the impression of our work.

---

### Author Rebuttal · Authors · 2023-08-10

We would like to thank all reviewers for their valuable and insightful comments. Attached to this post is a single page pdf containing 3 figures: B1, B2, and B3. B1 is a diagnostic visualization showcasing VIDM’s ability to intelligently copy pixels from previous frames. B2 visualizes failure modes of an ablation of our method that allows loss to propagate to pixels containing hands. B3 compares reconstructions from VIDM against those from NeuralDiff, demonstrating VIDM’s superior ability to recover occluded pixels. We have posted replies to each reviewer's individual comments to address their specific concerns.

---

### Decision · Program_Chairs · 2023-09-21

**Decision:**

Accept (poster)

**Comment:**

The reviewers found this work on factored scene representation to be making a valuable contribution despite some shortcomings in the evaluation. The AC thinks it is likely to give rise to similar methods useful for pixel-based decision-making scenarios beyond robotics. This work is recommended for acceptance, provided the authors incorporate their fixes to the issues discussed during the rebuttal stage into the camera-ready version.